# Epidemiology-informed Network for Robust Rumor Detection

## Abstract

The rapid spread of rumors on social media has posed significant challenges to maintaining public trust and information integrity. Since an information cascade process is essentially a propagation tree, recent rumor detection models leverage graph neural networks to additionally capture information propagation patterns, thus outperforming text-only solutions. Given the variations in topics and social impact of the root node, different source information naturally has distinct outreach capabilities, resulting in different heights of propagation trees. This variation, however, impedes the data-driven design of existing graph-based rumor detectors. Given a shallow propagation tree with limited interactions, it is unlikely for graph-based approaches to capture sufficient cascading patterns, questioning their ability to handle less popular news or early detection needs. In contrast, a deep propagation tree is prone to noisy user responses, and this can in turn obfuscate the predictions. In this paper, we propose a novel Epidemiology-informed Network (EIN) that integrates epidemiological knowledge to enhance performance by overcoming data-driven methods' sensitivity to data quality. Meanwhile, to adapt epidemiology theory to rumor detection, it is expected that each user's stance toward the source information will be annotated. To bypass the costly and time-consuming human labeling process, we take advantage of large language models to generate stance labels, facilitating optimization objectives for learning epidemiology-informed representations. Our experimental results demonstrate that the proposed EIN not only outperforms state-of-the-art methods on real-world datasets but also exhibits enhanced robustness across varying tree depths. We release the code at https://anonymous.4open.science/r/EIN-0104.

## Keywords

First Principle-guided Machine Learning; Rumor Detection; Graph Representation

**ACM Reference Format:**
Anonymous Author(s). 2024. Epidemiology-informed Network for Robust Rumor Detection. In . ACM, New York, NY, USA, 10 pages. https://doi.org/10.1145/nnnnnnn.nnnnnnn

## 1 Introduction

The pervasive integration of social media into daily life has significantly improved access to information. However, this has also led to a concomitant rise in the dissemination of false and fabricated content, known as rumors. Rumors rapidly propagate through social networks, undermining the integrity of the digital ecosystem and diminishing the quality of user interactions [18]. Particularly concerning is the propagation of malicious rumors, which can mislead the public, disrupt individual and societal equilibrium.

In light of these challenges, the development of advanced rumor detection technologies becomes imperative to curb the swift proliferation of misinformation. Earlier approaches encode text representations from original posts through natural language processing techniques. These methods leverage deep architectures such

as Recurrent Neural Networks (RNNs) [10, 22], Convolution Neural Networks (CNNs) [1, 28], and Transformers [16, 28] to embed and learn the textual content. However, these methods only rely on the linguistic features of online messages, neglecting the structural patterns of their propagation dynamics that are of high value to rumor detection. Notably, social media content tends to propagate hierarchically, where the original post will attract interactions (e.g., replies and reposts) that stimulate cascading interactions, thus forming a propagation tree [13]. To better model such tree-structured data, there is a recent shift towards leveraging graph neural networks (GNNs) to additionally model the structural information of rumor propagation, so as to enhance the detection accuracy [2, 7, 24–26].

Although graph-based rumor detectors offer promising improvements by jointly utilizing rumors' textual features and propagation mechanisms, the use of more complex propagation trees instead of single social media posts introduces new challenges. On the one hand, in shallow propagation trees where interactions are scarce (e.g., information that is new or from less influential users), graph-based methods can hardly capture useful signals for predictions. On the other hand, content authored by social influencers or with popular topics commonly forms deeper trees, which inherently contain higher heterogeneity in user responses. Consequently, this leads to a mixture of stances within the user-generated content in the tree, bringing substantial noise to the data. Unfortunately, existing graph-based rumor detectors struggle to address both issues effectively. To empirically verify this limitation, we conduct a preliminary experiment using two representative graph-based rumor detectors, RAGCL [7] and ResGCN [31], on propagation trees with varying depths, which is shown in Figure 1. This result reveals that neither of them consistently excels across all scenarios, indicating a lack of robustness against the structural complexities inherent in different tree depths. For example, RAGCL performs better on shallow trees. Ideally, a reliable rumor detector should not only accurately identify rumors at early stages when there are few user responses, but also effectively counter the noise in more complex propagation trees.

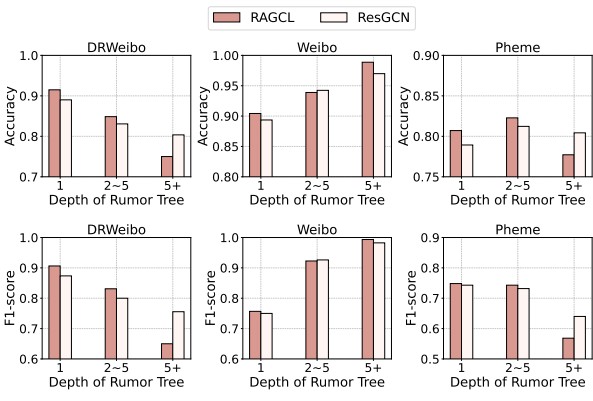

**Figure 1: Impact of tree depths on RAGCL and ResGCN illustrated with three real-world datasets, DRWeibo, Weibo and Pheme.**

The inherent vulnerability of rumor detection algorithms largely results from data quality, typical impediments in data-driven machine learning models. Therefore, we incorporate principled knowledge into the data-driven detection model, enabling it to maintain robust performance even under imperfect data conditions. Based on this, we propose a novel framework that enhances the resilience of rumor detectors by integrating epidemiological modeling. Epidemiological principles are widely used to describe rumor-spreading processes on social networks, offering a natural foundation for modeling rumor propagation dynamics [6, 8]. The transitions between states in epidemiology models, such as Susceptible and Infectious, align with the evolution of rumors in social media, from an initial Unknown state to eventual Support or Denial states. We leverage these dynamics to model the semantic progression of rumors using epidemiological principles. Concretely, we propose an Epidemiology-informed Encoder, which models the dynamics of three distinct states: Unknown, Support, and Denial. This encoder initializes and iteratively updates the embeddings for these states while optimizing the learning trajectory using state-specific labels.

Moreover, modeling and learning the three epidemiological states requires stance information between user responses. However, rumor detection datasets generally lack stance annotations, which results in lacking a ground truth for optimization. Annotating stance data is highly reliant on human expertise and is labor-intensive, making it impractical for large-scale applications. To address this limitation, we propose a novel approach that leverages a large language model (LLM) to label the stance of each post, that is using the world knowledge of LLM to guide the training of EIN effectively. Importantly, this stance generation by LLM occurs only during the training phase, serving exclusively as a means to enrich learning, not as a direct feature, which ensures operational efficiency and obviates the need for generation during the inference phase.

The main contributions of this paper are summarized as follows:

- We investigate the vulnerability of current graph-based rumor detectors across various depths of propagation trees, which is an unexplored issue in rumor detection.
- We propose a novel epidemiology-informed framework, which binds the first principle epidemiology theory with deep networks for enhancing rumor detectors. Besides, we utilize LLM-generated stances labels for optimizing epidemiology-informed representations of propagation trees without the need for manual annotations. Notably, this framework is compatible with any graph-based rumor detectors.
- We conduct comprehensive experiments on three real-world datasets with SOTA graph-based rumor detectors. The results demonstrate that our framework consistently outperforms existing methods across different depths of rumor propagation trees.

## 2 Preliminaries

### 2.1 Problem Definition

Rumor detection is formulated as a binary classification task. We consider a propagation tree $\mathcal{G}_i = \{V_i, E_i, X_i\}$, where $V_i = \{v_0, v_1, \ldots, v_n\}$ represents the nodes corresponding to posts within the event, and $v_0$ stands for the root post. $E_i$ represents the set of edges in the propagation tree. The node feature matrix $X_i$, defined as $[x_0, x_1, \ldots, x_n]^T$,

is extracted from the content of the posts using Word2Vec embeddings with the size of $h$, similar to the method described in [7].
**Problem Formalization.** The main task of this work is to develop a predictive model, that maps each root post $v_0^{(i)}$ with its graph-based structure $\mathcal{G}_i$ and textual features to a binary label $y_i \in \{0, 1\}$, representing non-rumor or rumor.

### 2.2 Epidemiological Transmission Model in Rumor Detection

The propagation of rumors has been widely described by epidemiology models, such as the Susceptible-Infectious (SI) and Susceptible-Infectious-Recovered (SIR) models [6, 32], which model the spread of epidemics through networks of human contact. Unlike typical epidemiological spread, rumors disseminate via hierarchical structures of propagation trees consisting of a root post and associated responsive posts, exhibiting distinct dynamics. As rumor detection concerns the truthfulness of the original post (i.e., the root of the tree), we specifically focus on how the overall population perceives the root post with different responsive posts.

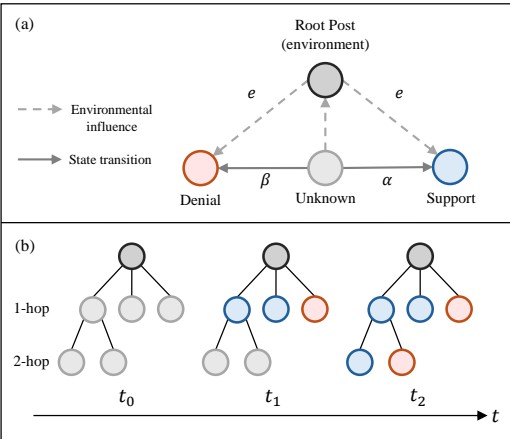

**Figure 2: Illustration of the eUSD model. (a) depicts the transition from the Unknown state to either Support or Denial, influenced by the root post. (b) presents a toy example of the rumor propagation process within a propagation tree.**

To more accurately capture these dynamics, we take advantage of an environmental transmission model [4, 29], which allows us to model the propagation process within a propagation tree through an environmental Susceptible-Infectious relationship. In a nutshell, when new users generate new posts in the propagation tree, their states relative to the root post transit from an Unknown state to either Support or Denial over time, as depicted in Figure 2. We introduce the environmental Unknown-Support-Denial (eUSD) model:

$$\begin{cases} \dfrac{dU}{dt} = -\alpha U e - \beta U e, \\ \dfrac{dS}{dt} = \alpha U e, \\ \dfrac{dD}{dt} = \beta U e, \end{cases} \tag{1}$$

where $t$ is the discretized propagation stage (i.e. the depth of propagation tree), $U, S, D$ are the number of individuals of Unknown,

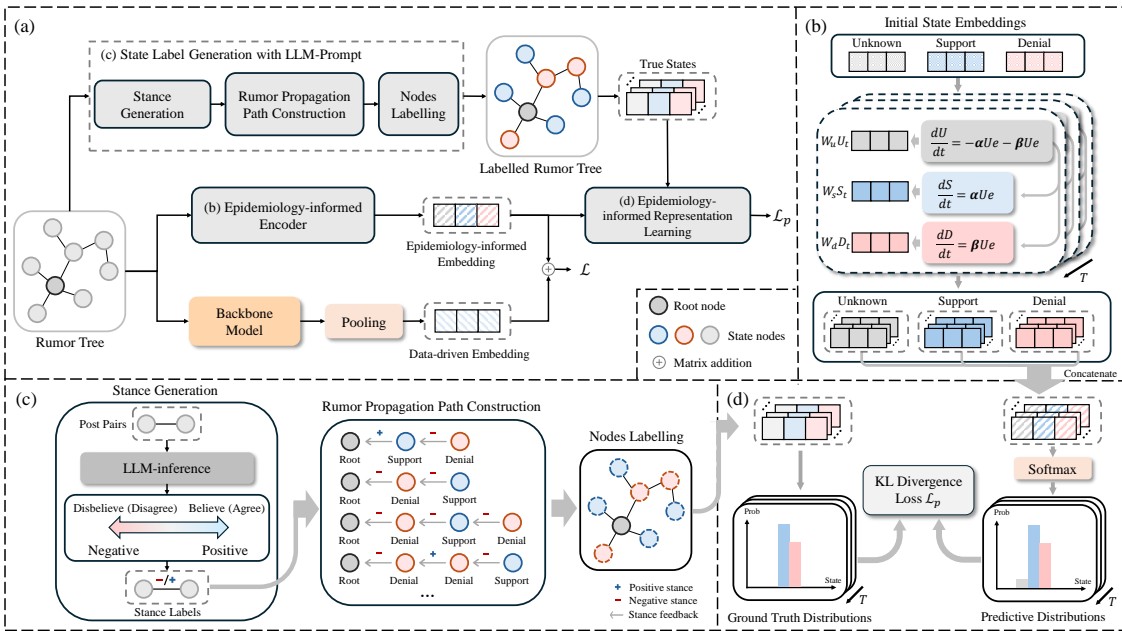

**Figure 3: The overview of Epidemiology-informed Network. (a) is the workflow of the rumor detector combining epidemiology-informed embedding and the data-driven embedding. EIN consists of three main modules: (b) Epidemiology-informed Encoder. (c) State Label Generation with LLM-Prompt. (d) Epidemiology-informed Representation Learning.**

Support and Denial states, respectively. $\alpha$ and $\beta$ are the transition probabilities from Unknown to Support and Denial states, respectively. $e$ is the environment influence rate, which in our context depends on the root post (e.g., its author, topic, publication time, etc.). This model demonstrates that the rate at which individuals transition to Support or Denial states is influenced by the root posts (environmental factor) and the number of individuals in the Unknown state. As $e$ is a universal scaling factor, without loss of generality, we set $e=1$ and mainly focus on $\alpha$ and $\beta$ in our work. In Section 3.4, we will discuss the differences between eUSD and traditional epidemiology formulation, and further compare their effectiveness via ablation study in Section 4.4.

## 3 Methodology

In this section, we introduce the Epidemiology-informed Network (EIN), a principled framework designed to integrate seamlessly with various graph-based rumor detectors. As shown in Figure 3, the EIN framework is composed of three main modules: the Epidemiology-informed Encoder, the State Label Generation with LLM-Prompt, and the Epidemiology-informed Representation Learning. Detailed descriptions of each component are provided below.

### 3.1 Epidemiology-informed Encoder

Current graph-based rumor detection architectures benefit from capturing information in both the propagation and dispersion directions of a propagation tree through graph aggregation layers. However, these data-driven models struggle with limited data availability or when the propagation structure introduces excessive noise. To overcome these challenges, we employ the eUSD model

to encode state representations, aiming to enrich the data-driven model by incorporating the principle of rumor propagation dynamics as an auxiliary feature. We first initialize the state embeddings of Unknown, Support and Denial $U_{t_0} \in \mathbb{R}^h, S_{t_0} \in \mathbb{R}^h, D_{t_0} \in \mathbb{R}^h$:

$$U_{t_0} = b_u \hat{u}, \ \ S_{t_0} = b_s, \ \ D_{t_0} = b_d, \tag{2}$$

where $\hat{u}$ is the number of nodes in a tree, $b_u$, $b_s$ and $b_d$ are the learnable embeddings for Unknown, Support and Denial states, respectively. Since the initialization of $U$ is the only embedding based on the number of nodes in a tree, while others are randomly initialized, we aim to imbue $U_{t_0}$ with specific rumor propagation properties at the outset. To achieve this, we incorporate the discretized dynamics of $U$ during its initialization. We define $b_u = W_{u_0} - \alpha W_{u_0} - \beta W_{u_0}$, where $W_{u_0}$ is a learnable parameter.

In Section 2.2, we formalize the process of rumor propagation using a continuous formulation. However, encoding a continuous eUSD model into a rumor detector is challenging and requires more computational resources, similar to dynamic systems in other applications. [9, 15]. Consequently, we shift our focus to a discrete modeling approach, which is particularly well-suited for analyzing the dynamics of rumor propagation. Concretely, we use a discretized forward difference quotient to approximate Eq. (1). We first define the update process of the Unknown state embedding $U_t \in \mathbb{R}^h$:

$$U_{t+1} = U_t - \alpha U_t - \beta U_t, \tag{3}$$

where $\alpha, \beta \in (0, 1)$ are learnable parameters. Then, $S_t \in \mathbb{R}^h$ and $D_t \in \mathbb{R}^h$ can be updated as follows:

$$\begin{aligned} S_{t+1} &= W_s(S_t + \alpha W_u U_t), \\ D_{t+1} &= W_d(D_t + \beta W_u U_t), \end{aligned} \tag{4}$$

where $W_u$, $W_s$ and $W_d$ are used to capture and learn the dynamics of $U$, $S$ and $D$, respectively. Then we concatenate $U_T$, $S_T$ and $D_T$ as an epidemiology-informed representation that encodes the information propagation dynamics at the entire tree level:

$$x_g = W_x([U_T; S_T; D_T]), \tag{5}$$

where $T$ is the depth of the propagation tree, $W_x$ is the parameter for learning the combination of three states and resizing $x_g$.

We input the node feature $X$ into an arbitrary graph-based rumor detector $f$, obtaining the tree-level representation $x_f$ after the graph pooling:

$$x_f = \text{Pooling}(f(X)), \tag{6}$$

where $x_f$ is a fully data-driven embedding. Given the epidemiology-informed embedding $x_g$, it can be integrated with $x_f$ as follows:

$$\hat{y} = \text{softmax}(W_l(x_f + x_g)), \tag{7}$$

where $W_l$ is the linear layer for decoding and $\hat{y}$ is the distribution over the two classes.

## 3.2 State Label Generation with LLM-Prompt

Learning state representations necessitates the availability of annotated state labels for responsive posts in each propagation tree, such that an additional optimization goal can be set to ensure that the learned $U_t$, $S_t$ and $D_t$ accurately reflect the true information dynamics at $t$. However, the prohibitive cost of performing human annotation on each post's stance inevitably calls for a more efficient alternative. The expansive parameterization of powerful large language models (LLMs) equips them to adeptly discern the stance between two given sentences. As such, we leverage LLMs to generate stance labels. Then, we can generate the state labels of nodes based on the stance labels. These state labels represent the attitude of a responsive post towards the root post.

---

**Algorithm 1:** State label generation with LLM-Prompt.

**Input:** A propagation tree $\mathcal{G}$ with a set of nodes $V$ including a root node $v_0$ and other nodes.
**Output:** A expanded propagation tree $\mathcal{G}'$ with state labels $C[v] \in \{0, 1\}$ for all nodes in $V$ except $r$

1 Initialize an array $C$ to store the state of each node;
2 **for** $v \in V \setminus \{v_0\}$ **do**
3      $\text{stance}(v) \leftarrow \text{LLM}(s(\text{parent}(v)), s(v))$;
         // $\text{stance}(v) \in \{0, 1\}$
4      **if** $\text{parent}(v) = v_0$ **then**
5          **if** $\text{stance}(v) = 0$ **then**
6              $C[v] \leftarrow 0$;
7          **else**
8              $C[v] \leftarrow 1$;
9      **else**
10          $C[v] \leftarrow C[\text{parent}(v)] \oplus \text{stance}(v)$;
11 $\mathcal{G}' = \mathcal{G} \cup \{C\}$
12 **return** $\mathcal{G}'$.

---

The process of state label generation is demonstrated in Algorithm 1, where $v \in V \setminus \{v_0\}$ denotes the set of nodes excluding

the root node, $\text{LLM}(\cdot)$ is used to represent the call to the language model, the prompt setting of LLM is shown in Appendix A. $s(v)$ represents the sentence at node $v$, $\text{parent}(v)$ denotes the parent of node $v$, $C[v]$ denotes the state label of node $v$, $\oplus$ is the exclusive OR operation. Note that there are two categories of stance labels, $\text{stance}(v) \in \{0, 1\}$, where 0 represents "Positive" and 1 represents "Negative". Similarly, there are two categories of state labels: 0 for "Support" and 1 for "Denial". In line 3, for each node except the root, we input the original content of the text sentence and its parent's text sentence into an LLM to generate the stance label $\text{stance}(v)$ for node $v$. In lines 4 to 8, if the parent of node $v$ is the root node, we directly assign the state label to node $v$ based on its stance label. Conversely, as specified in line 10, if the parent of node $v$ is not the root node, the state label of node $v$ is determined by its parent's state label in conjunction with its own stance label. Finally, we expand the original propagation tree with the generated state labels $C$, transforming it into an updated propagation tree $\mathcal{G}'$ in line 11.

## 3.3 Epidemiology-informed Representation Learning

The Epidemiology-informed Encoder refrains from directly incorporating the generated state labels into the modeling process. This is motivated by the significant computational burden and time constraints associated with employing large language models (LLMs) to generate additional state labels during the inference phase. Instead, we adopt a strategy where the encoder simulates the dynamic process with randomly initialized embeddings. To enhance the realism and efficacy of this simulation, the generated state labels are used indirectly to guide the learning process. The ultimate objective is to refine the encoder such that its output distribution closely approximates the distribution of the generated state labels, which can be defined as:

$$\mathcal{L}_p = \sum_{t=1}^{T} \left( \text{KL}(p_{u,t} \| \hat{p}_{u,t}) + \text{KL}(p_{s,t} \| \hat{p}_{s,t}) + \text{KL}(p_{d,t} \| \hat{p}_{d,t}) \right), \tag{8}$$

where $p_{u,t}, p_{s,t}, p_{d,t}$ represent the distributions of generated state labels at stage $t$. Given the state label of Unknown $C^u$ in the propagation tree, $p_{u,t} = \frac{|C^u|}{n-1}$ ($n$ is the number of nodes in the tree), the distribution of the other two states can be computed analogously. $\hat{p}_{u,t}, \hat{p}_{s,t}, \hat{p}_{d,t}$ denote the distributions transformed from $U_{1:T}, S_{1:T}, D_{1:T}$ and normalized using the Softmax function. $\text{KL}(p \| \hat{p})$ represents the Kullback–Leibler divergence between the distributions of generated state labels and the distributions learned from the encoder. We unify the joint loss function of EIN:

$$\mathcal{L} = \mathcal{L}_r + \lambda \mathcal{L}_p, \tag{9}$$

where $\mathcal{L}_r$ is the cross-entropy loss as the rumor detection loss. Given that the stance labels generated by LLM are not entirely accurate, we assign a coefficient $\lambda$ to control the effect of epidemiology-informed representation learning.

## 3.4 Discussion on Epidemiology Model

We define a transmission model that uniquely characterizes the propagation of rumors based on the epidemiology model eUSD, which is a generalized version of Unknown-Support-Denial (USD), a commonly used epidemiology model. We discuss the rationale of

building our model upon eUSD rather than USD by starting with USD's formulation:

$$
\begin{cases}
\dfrac{dU}{dt} = -\alpha US - \beta UD, \\[2mm]
\dfrac{dS}{dt} = \alpha US, \\[2mm]
\dfrac{dD}{dt} = \beta UD.
\end{cases}
\tag{10}
$$

Compared with Eq. (1), Eq. (10) mainly focuses on the mutual interactions of $U$, $S$, and $D$ over time without the intervention of environmental factor $e$. That is to say, the transition from Unknown state to Support/Denial additionally depends on the overall densities of $S$ and $D$. However, this only holds if all users' own opinions are driven by the entire population's stances, which is an impractical assumption in reality. In social media, most posts in a propagation tree are a user's direct reaction towards the root post. As such, with eUSD's formulation in Eq. (1), a node in the Unknown state at time $t$ is not predominantly influenced by the number of nodes in Support or Denial states across the entire tree, but rather by the environment set by the root post. Though it is unrealistic to assume that all users globally affect an individual in USD, a user's stance does tend to be locally affected by surrounding users (e.g., connected friends) in the propagation tree. Thus, the fusion of the eUSD-based and graph-based representations in Eq. (7) effectively factors in the joint effect from the root post and the locally connected nodes in the rumor detection task.

To empirically validate and enhance the EIN model utilizing the principles of our defined eUSD model, we establish a variant by introducing a regular USD model to describe the dynamics of rumor propagation. We also use a discretized forward difference quotient to encode the state embeddings and incorporate them into the rumor detector as a variant of EIN. We conduct the ablation study to compare the effectiveness of both EINs in Section 4.4, the experimental results indicate that the EIN utilizing eUSD model outperforms the EIN utilizing the regular USD model, indirectly suggesting that the modifications incorporated into the eUSD model are more reasonable and effective in modeling rumor propagation dynamics.

## 4 Experiments

To verify the effectiveness of EIN, we conduct extensive experiments on three real-world datasets and answer the following research questions (RQs):

- **RQ1**: How does the EIN perform compared to state-of-the-art models (SOTAs) in graph-based rumor detection?
- **RQ2**: How does EIN enhance the backbone models compared to the vanilla version?
- **RQ3**: How does the EIN perform on the samples with various tree depths?
- **RQ4**: What is the effect of the components of EIN?
- **RQ5**: What is the impact of hyperparameter settings on EIN?

## 4.1 Experimental Settings

*4.1.1 **Datasets***. We conduct our experiments utilizing three publicly available real-world datasets tailored for graph-based rumor detection: DRWeibo [7], Weibo [21], and Pheme [33]. Each dataset

**Table 1: Statistics of rumor detection datasets**

| Statistics | DRWeibo | Weibo | Pheme |
|---|---|---|---|
| # source posts | 6037 | 4664 | 5748 |
| # non-rumors | 3185 | 2351 | 3654 |
| # rumors | 2852 | 2313 | 2094 |
| language | zh | zh | en |

comprises the texts of root post and responsive posts, and the hierarchical structure of the user responses. Notably, the DRWeibo and Weibo datasets contain content in Chinese (zh), whereas the Pheme dataset consists of content in English (en). All datasets are used for binary classification tasks. The descriptive statistics of these datasets are presented in Table 1.

*4.1.2 **Baselines***. We compare EIN with 10 rumor detection baselines belonging to two categories.

**Graph Classification:**

- **GCN** [17]: A neural network architecture that operates on graph-structured data by applying convolutional operations to learn node features based on their local neighborhoods.
- **GIN** [30]: A graph neural network that focuses on distinguishing graph structures by learning powerful node representations through a sum aggregation function.
- **KAGNN** [3]: A method that combines Kolmogorov-Arnold Network and GIN, providing better modeling of complex systems.
- **ResGCN** [31]: An advanced setting of GCN, which can produce better robustness and effectiveness.

**Graph-based Rumor Detection:**

- **LeRuD** [20]: A prompt strategy for generating detection results using large language model (LLM) inference. Due to the limitations of the GPT implementation and fair comparison with our method, we utilize Gemma2-9B as the LLM.
- **BiGCN** [2]: A graph-based model that encodes both significant rumor propagation and dispersion to capture the global structure of the propagation tree.
- **UDGCN**: A variant of BiGCN that directly leverages single encoder for modeling undirected propagation tree.
- **GACL** [25]: A graph-based method leverages adversarial and contrastive learning to encode the global propagation structure.
- **GARD** [26]: A rumor detection model incorporates self-supervised semantic evolution information to enhance representation of event propagation.
- **RAGCL** [7]: A graph-based rumor detection framework that integrates graph contrastive learning by using node centrality to augment the views of data.

*4.1.3 **Evaluation Metrics and Implementation Details***. We split the dataset into training, validation, and test sets with a 6:2:2 ratio, and we adopt three common evaluation metrics for the rumor detection task: Accuracy (Acc.), ROC-AUC, and F1-score (F1). To ensure a fair comparison, each model is trained five times, and we report the average results along with the standard deviations.

We implement the proposed EIN using PyTorch and conduct the experiments on an RTX 4090 GPU. Following the setting of RAGCL, we represent the node-wise text content using Word2Vec

**Table 2: Overall performance comparison of models across three datasets. The best results are highlighted in bold, and the second-best results are underlined.**

| Model | DRWeibo | | | Weibo | | | Pheme | | |
|---|---|---|---|---|---|---|---|---|---|
| | ACC (%) | AUC (%) | F1 (%) | ACC (%) | AUC (%) | F1 (%) | ACC (%) | AUC (%) | F1 (%) |
| GCN | 85.28±2.25 | 84.88±2.52 | 83.62±3.55 | 91.86±1.93 | 91.93±1.81 | 91.89±1.86 | 79.13±1.72 | 76.88±3.34 | 69.81±5.15 |
| GIN | 83.79±2.04 | 84.00±1.78 | 83.81±1.41 | 91.60±1.15 | 91.57±1.21 | 91.84±0.98 | 79.34±1.20 | 77.59±1.44 | 71.17±2.03 |
| KAGNN | 85.56±0.68 | 85.41±0.74 | 84.46±1.22 | 92.41±1.55 | 92.46±1.54 | 92.33±1.87 | 79.90±1.23 | 78.28±2.17 | 72.03±2.71 |
| ResGCN | 86.23±1.11 | 86.17±1.09 | 85.21±1.51 | 93.98±0.90 | 94.00±0.87 | 94.04±0.83 | 79.86±0.87 | 78.61±1.03 | 72.52±1.86 |
| LeRuD | 70.00±0.90 | 70.39±0.94 | 71.18±1.03 | 70.38±0.72 | 70.41±0.76 | 71.80±0.46 | 44.13±0.77 | 47.10±0.93 | 42.47±0.95 |
| UDGCN | 80.08±3.23 | 80.50±2.92 | 81.18±2.29 | 90.91±1.17 | 90.90±2.14 | 91.55±2.09 | 80.89±1.12 | 77.87±1.87 | 71.52±2.94 |
| BiGCN | 84.64±3.44 | 84.59±3.11 | 84.55±2.10 | 93.65±0.90 | 93.62±0.89 | 93.92±0.78 | 82.00±1.28 | 79.76±3.38 | 73.84±4.64 |
| GACL | 84.93±1.78 | 84.98±1.90 | 84.39±2.96 | 93.98±0.96 | 93.96±0.99 | 94.12±0.82 | 79.11±0.46 | 78.22±0.62 | 72.01±0.78 |
| RAGCL | 87.41±0.62 | 87.18±0.71 | 86.23±1.12 | 93.95±0.84 | 93.95±0.84 | 93.95±0.95 | 81.71±0.52 | 79.81±1.69 | 74.05±2.55 |
| **EIN (ours)** | **88.01±0.64** | **87.97±0.62** | **87.23±0.64** | **95.16±0.59** | **95.13±0.60** | **95.19±0.68** | **82.74±0.77** | **81.78±1.30** | **76.45±1.66** |

**Table 3: Comparison of backbones with different frameworks and their improvements. The best results are highlighted in bold.**

| Model | | DRWeibo | | | Weibo | | | Pheme | | |
|---|---|---|---|---|---|---|---|---|---|---|
| | | ACC (%) | AUC (%) | F1 (%) | ACC (%) | AUC (%) | F1 (%) | ACC (%) | AUC (%) | F1 (%) |
| BiGCN | vanilla | 84.64±3.44 | 84.59±3.11 | 84.55±2.10 | 93.65±0.90 | 93.62±0.89 | 93.92±0.78 | 82.00±1.28 | 79.76±3.38 | 73.84±4.64 |
| | + RAGCL | 81.81±3.48 | 82.17±3.13 | 82.50±2.41 | 92.26±1.42 | 92.22±1.39 | 92.71±1.27 | 81.71±0.52 | 79.81±1.69 | 74.05±2.55 |
| | RAGCL-improv. | -3.34% | -2.86% | -2.42% | -1.49% | -1.50% | -1.29% | -0.36% | +0.06% | +0.28 % |
| | **+ EIN** | **86.93±1.38** | **86.82±1.27** | **86.31±0.78** | **94.62±0.64** | **94.64±0.62** | **94.64±0.60** | **82.74±0.77** | **81.78±1.30** | **76.45±1.66** |
| | EIN-improv. | +2.71% | +2.64% | +2.09% | +1.03% | +1.10% | +0.77% | +0.98% | +2.60% | +3.64% |
| ResGCN | vanilla | 86.23±1.11 | 86.17±1.09 | 85.21±1.51 | 93.98±0.90 | 94.00±0.87 | 94.04±0.83 | 79.86±0.87 | 78.61±1.03 | 72.52±1.86 |
| | + RAGCL | 87.41±0.62 | 87.18±0.71 | 86.23±1.12 | 93.95±0.84 | 93.95±0.84 | 93.95±0.95 | 80.31±0.66 | **78.82±1.02** | **72.78±1.75** |
| | RAGCL-improv. | +1.37% | +1.18% | +1.20% | -0.02% | -0.05% | -0.10% | +0.57% | +0.26% | +0.37% |
| | **+ EIN** | **88.01±0.64** | **87.97±0.62** | **87.23±0.64** | **95.16±0.59** | **95.13±0.60** | **95.19±0.68** | **80.32±0.81** | 78.80±0.91 | 72.76±1.62 |
| | EIN-improv. | +2.06% | +2.10% | +2.37% | +1.26% | +1.20% | +1.22% | +0.57% | +0.24% | +0.33% |

embeddings, and we maintain the same hyperparameters for the backbone models as in RAGCL, including a batch size to 128, a learning rate to 0.0005 for DRWeibo and Weibo, and 0.0001 for Pheme, and embedding size to 200 [7]. Furthermore, Section 4.5 reports the analysis of influences on different initialization of parameters $\alpha$, $\beta$ and the coefficient of epidemiology-informed representation learning $\lambda$. We use Gemma 2-9B [27] as the LLM for stance generation, the temperature of LLM is set to 0.2.

## 4.2 Overall Comparison

To address RQ1, we compare EIN against two groups of baseline models, with the results summarized in Table 2. Moreover, to evaluate the effectiveness and generalizability of EIN (RQ2), we compare the vanilla backbone models, RAGCL-enhanced backbones, and EIN-enhanced backbones, which is presented in Table 3. Based on these results, we observe the following:

**State-of-the-Art performance across baselines.** Table 2 demonstrates the effectiveness of EIN, which achieves the best accuracy, ROC-AUC, and F1-score comparing the baselines from both graph classifiers and graph-based rumor detectors across three datasets. Notably, it outperforms the most recent SOTA graph-based rumor detection models, GARD and RAGCL. Furthermore, EIN leverages pseudo labels generated by LLM and incorporates an epidemiology-informed architecture, which enhances tree-level representations to boost overall performance. We also evaluate the performance of LeRUD, a strategy employing LLM inference for rumor detection. However, its efficacy is constrained by the capabilities and

scale of the underlying LLM. In contrast, EIN demonstrates robust performance that is not limited by these factors.

**Generalizability of EIN.** EIN is designed as a versatile, plug-and-play framework applicable across various vanilla graph-based rumor detection models. We integrate EIN with two representative models, BiGCN and ResGCN, similarly to the implementation in RAGCL, to demonstrate its adaptability. As shown in Table 3, EIN consistently enhances the performance of both BiGCN and ResGCN across three datasets. This improvement underscores the utility of our epidemiology-informed representation learning, which proves adaptable to diverse backbones. While RAGCL incorporates contrastive learning loss and feature enhancement, it occasionally underperforms, highlighting the superior adaptability and effectiveness of the proposed EIN.

## 4.3 Robustness on Different Tree Depths

To address RQ3, we conduct experiments to assess the robustness of the EIN across samples distinguished by their tree depths. Specifically, we evaluate the performance of EIN on propagation trees with three various depth categories: depth = 1, depths ranging from 2 to 5, and depth ≥ 5. The distribution of these tree depths within our dataset is detailed in Table 4, and Figure 4 illustrates the impact of these varying tree depths on the performance of EIN, RAGCL and ResGCN, providing insights into the adaptability and effectiveness of our approach under diverse structural conditions.

From the experimental results, we draw the following conclusions: (1) In instances where the propagation tree has shallow depth,

**Table 4: The distribution of tree depths in test samples.**

| Dataset | Depth=1 | Depth=2~5 | Depth>5 |
|---------|---------|-----------|---------|
| DRWeibo | 53.65% | 41.69% | 4.65% |
| Weibo | 10.08% | 61.41% | 28.51% |
| Pheme | 34.32% | 49.65% | 16.02% |

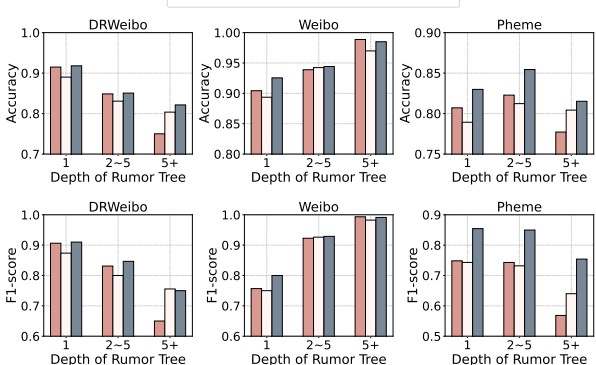

**Figure 4: Impact of tree depths to model performances.**

limited information is available, making it challenging for ResGCN to effectively capture and learn features. While RAGCL attempts to mitigate this issue by leveraging node centrality to augment data views, it struggles with trees of deeper depths, which tend to contain redundant noise. However, the proposed EIN utilizes epidemiology-informed representation to more adeptly capture the dynamics of rumor propagation, thereby enhancing performance across various tree depths. (2) As shown in Table 4, when the depth $\geq$ 5 in the DRWeibo dataset and depth=1 in the Weibo dataset, there is a paucity of samples. Under such conditions of data sparsity, baseline methods exhibit limited effectiveness. In contrast, our EIN maintains robust performance even with fewer samples, demonstrating superior adaptability and efficiency in handling data-scarce scenarios.

### 4.4 Ablation Study

To rigorously evaluate the individual contributions of the components within the EIN and answer RQ4, we conduct an ablation study contrasting the performance of EIN with two of its variants. Specifically, we explored: (1) EIN w/o ERL, which omits the epidemiology-informed representation learning component; and (2) EIN-reg, which incorporates a regular transmission model as described in Section 3.4.

The results of this ablation study are depicted in Figure 5. The results demonstrate that EIN consistently surpasses its variants, underscoring the critical roles of both the epidemiology-informed representation learning and the refined environmental transmission model in boosting performance. Notably, the performance disparities across datasets reveal that the epidemiology-informed representation learning markedly benefits the Pheme dataset, whereas the precise and adaptive transmission model significantly enhances outcomes on both DRweibo and Weibo.

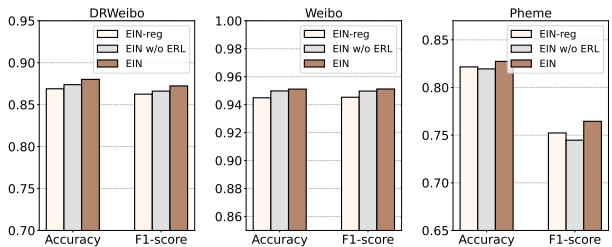

**Figure 5: Performance comparison of ablation study.**

### 4.5 Hyperparameter Analysis

To answer RQ5, we conduct sensitivity analysis on the hyperparameters of EIN, focusing on the initialization of infectious rates, denoted $\alpha$ and $\beta$, and the coefficient of epidemiology-informed representation learning $\lambda$. The findings from this analysis are presented in Table 5 and Figure 6.

*4.5.1 Effect of the Initialization of Infectious Rates.* We investigate various initializations of the infectious rates $\alpha$ and $\beta$, which respectively represent the probability of transitions from the Unknown to a Support state and from Unknown to Denial. In the sensitivity analysis, these rates are initialized at values of 0, 0.5, 1, and a random value between 0 to 1, with optimization occurring during subsequent training phases. The result shown in Table 5 reveals that optimal performance on the DRWeibo dataset is achieved when both rates are initialized at 0, and the rates at 0.5 result in optimal performance on both the Weibo and Pheme datasets.

**Table 5: Effect of various infectious rates $\alpha$ and $\beta$ initialization on model performance (%).**

| $\alpha_0, \beta_0$ | DRWeibo | | | Weibo | | | Pheme | | |
|---|---|---|---|---|---|---|---|---|---|
| | ACC | AUC | F1 | ACC | AUC | F1 | ACC | AUC | F1 |
| 0.0 | **88.01** | **87.97** | **87.23** | 94.28 | 94.25 | 94.29 | 82.42 | 80.22 | 74.53 |
| 0.5 | 87.66 | 87.69 | 87.30 | **95.16** | **95.13** | **95.19** | **82.74** | **81.78** | **76.45** |
| 1.0 | 87.99 | 87.90 | 87.20 | 95.03 | 95.01 | 95.06 | 82.54 | 80.69 | 75.14 |
| Rd. | 87.67 | 87.55 | 86.83 | 94.34 | 94.34 | 94.31 | 82.00 | 79.96 | 74.17 |

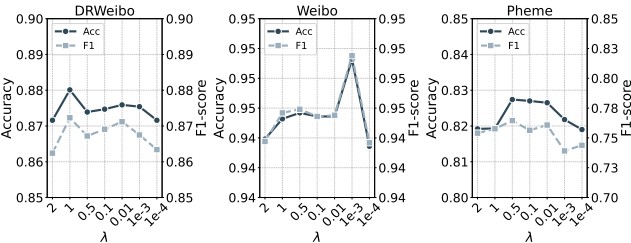

**Figure 6: Effect of epidemiology-informed representation learning coefficient $\lambda$.**

*4.5.2 Effect of Epidemiology-informed Representation Learning Coefficient.* The coefficient $\lambda$ controls the strength of the loss attributed to epidemiology-informed representation learning. To assess the sensitivity of this parameter, we select representative values and evaluate their impact. According to the results illustrated in Figure 6, setting $\lambda$ to values of 1, 0.001, and 0.5 yields optimal performances on the DRWeibo, Weibo, and Pheme datasets, respectively. Notably, $\lambda$ demonstrates low sensitivity on the DRWeibo and Pheme datasets.

## 4.6 Case Study

We conduct a case study to evaluate the capabilities of our EIN in managing scenarios characterized by sparse data and complex rumor propagation patterns within graph-based rumor detection. The results of the case study are shown in Figure 7. We randomly selected two non-rumor and two rumor samples from the DRWeibo and Weibo datasets, examining their temporal evolution in terms of Support and Denial state distributions.

As shown in Figure 7a, the initial distribution for sample 494 (non-rumor) shows a significantly large Support state area and a small Denial state area. In contrast, for sample 814 (rumor), the Support state area remains consistently small, whereas the Denial state area remains large throughout the observed period. Additionally, in Figure 7b, sample 748 (non-rumor) exhibits a larger overall Support state area compared to sample 474 (rumor), and a correspondingly smaller Denial state area, indicating distinctive patterns in rumor and non-rumor propagation dynamics.

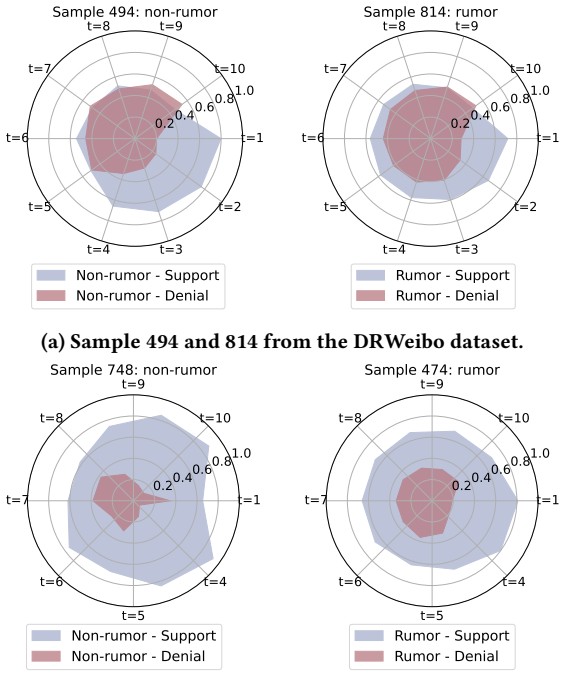

(a) Sample 494 and 814 from the DRWeibo dataset.

(b) Sample 748 and 474 from the Weibo dataset.

**Figure 7: Case study on DRWeibo and Weibo datasets.**

## 5 Related Work

### 5.1 Graph-based Rumor Detection

Recent graph-based rumor detection methods leverage graph theory and graph neural networks to adeptly capture the correlations between user responses and patterns of information propagation, enabling precise differentiation between rumors and genuine information in rumor propagation trees [2, 7, 24–26]. Most of the current methods rely on BiGCN to learn the propagation patterns from both top-down and bottom-up directions of propagation trees [2]. For instance, GARD utilizes a bi-directional graph autoencoder to learn semantic evolvement information [26]. DDGCN generates

duel-dynamic knowledge graphs to learn the dynamic event representations of rumor propagations [24]. Some remarkable works apply graph contrastive learning to enhance the views of data by using node centrality or sample variance [7, 25]. Despite these advancements, the complexity of propagation trees can vary significantly among samples. Some trees with minimal comments may lack substantial information, while others, rich in data, can be overly complex and redundant. Existing methods struggle to address both scenarios effectively. In response, we propose a novel approach that integrates inherent knowledge of information propagation dynamics, aimed at improving the adaptability and accuracy of rumor detection across diverse data complexities.

Moreover, FSNet incorporates the rumor propagation path in its modeling framework, which requires the use of authentic stance labels for supervision [19]. However, the task of annotating user responses with stance labels presents significant challenges and demands considerable time in most real-world settings [21, 33]. Our proposed method leverages existing large-scale language models to generate pseudo-stance labels, enabling the effective capture of user stance features, which can be effectively utilized in our physics-guided component.

### 5.2 First Principle-guided Machine Learning

Data-driven machine learning approaches often fail to capture the complex dynamics present in real-world applications, primarily due to constraints imposed by data quality and their inherently opaque structures, which tend to overlook essential physical principles. To overcome these challenges, physics-guided machine learning effectively combines physical laws with the adaptability of data-driven strategies. These methods have been successfully applied in a variety of real-world scenarios, including epidemiology, traffic flow, weather forecasting, and air quality prediction [5, 12, 14, 15, 23].

To address the limitations posed by scarce data, prevalent strategies include modifying the loss function to embed physical laws as constraints during the optimization process or augmenting the dataset with simulations derived from physics-based models [5, 11, 14, 23]. Alternatively, some studies have explored the integration of active learning techniques with physics-guided machine learning methods or adjustments to the architecture of deep learning models [12, 15]. These modifications aim to develop more robust approaches that effectively reduce redundancy in the data.

## 6 Conclusion

In this paper, we propose the Epidemiology-informed Network to enhance the robustness of graph-based rumor detection across various propagation tree structures. We develop an Epidemiology-informed Encoder to model state embeddings and seamlessly plug it into any graph-based rumor detector. Additionally, we employ large language models to generate stance labels, which are then utilized to optimize the epidemiology-informed embeddings. The effectiveness of the proposed EIN framework is validated through a series of experiments conducted on three real-world datasets. The results demonstrate that the EIN achieves state-of-the-art performance while maintaining robustness across propagation trees of varying depths.

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

# A Prompts of Stance Generation

In this section, we introduce the prompt settings used to generate stance labels for each responsive post within a propagation tree. To determine the stance label of a responsive post relative to the root post, it is first necessary to generate the stance label of the post in relation to its parent post. This process involves two steps of generation. If the parent post is the root post, the following prompt is utilized:

> - **Source post:** '*source_sentence*'
> - **Responsive post:** '*response_sentence*'
> - Based on the content of the response comment, determine its attitude towards the source post and choose one of the following options: The response comment believes the source post: 0, The response comment does not believe (or doubts) the source post: 1. If the response comment only contains '@' someone(s) without any other content, then you can consider that the response is believing the source post. You only need to select one label from the options above as the final result, no additional text is required.

If the parent post is not the root post, the following prompt is utilized:

> - **Source post:** '*source_sentence*'
> - **Responsive post:** '*response_sentence*'
> - Based on the content of the response sentence, determine its attitude towards the source sentence and choose one of the following options: The response sentence agrees with the source sentence: 0, The response sentence disagrees (or doubts) the source sentence:1. If the response sentence only contains '@' someone(s) without any other content, then you can consider that the response is agreeing to the source sentence. You only need to select one label from the options above as the final result, no additional text is required.

Given a source post and its corresponding responsive post, we assign a stance label to the responsive post, categorizing it as either positive (label: 0) or negative (label: 1). Once the stance label is determined, we then follow the steps outlined in Algorithm 1 to generate the state label, which reflects the responsive post's attitude toward the root post.