# OpenReview forum: "Epidemiology-informed Network for Robust Rumor Detection"
_ACM.org/TheWebConf/2025/Conference — WWW 2025 Poster_

### Official Review · Reviewer_BXVP · 2024-11-08

**Novelty:** 6
**Technical Quality:** 6

**Review:**

Pros:

1.	Introducing epidemiology theory into the method design for rumor detection is a novel and interesting research contribution.
2.	The experiments and analyses are extensive and thorough.
3.	This paper is well-organized, and the writing is easy to follow. The visualization is nicely presented.

Cons:

1.	A bit more elaboration is needed to illustrate the motivation for adopting epidemiology-informed approaches to improve the robustness of graph-based rumor detection methods. How is explicitly modeling three states (Unknown, Support, and Denial) able to handle propagation trees of various depths better? I think the method’s effectiveness has been confirmed empirically by extensive experiments; what I’m curious about is whether the authors have any intuitive explanations on why epidemiology-informed approaches are particularly effective.
2.	I wonder how good the quality of LLM-generated stance labels is. Have the authors considered measuring it with either LLM-judge or a small scale of human evaluation?

**Questions:**

See Cons

**Reviewer Confidence:**

3: The reviewer is confident but not certain that the evaluation is correct

**Scope:**

4: The work is relevant to the Web and to the track, and is of broad interest to the community

---

### Official Review · Reviewer_rr7D · 2024-11-12

**Novelty:** 4
**Technical Quality:** 4

**Review:**

The paper introduces an Epidemiology-Informed Network (EIN) model for rumor detection that leverages epidemiological models and large language models (LLMs) to generate classification labels. The approach simulates user attitude changes (e.g., supportive or oppositional) during rumor propagation, enhancing detection accuracy. By incorporating epidemiological knowledge to model dynamic shifts in user attitudes, the proposed EIN method employs LLMs to generate labels automatically, thereby reducing the need for manual annotation. This approach addresses the limitations of traditional Graph Neural Network-based rumor detection models, which often require high-quality data and struggle with instability across varying propagation tree depths.

The proposed method consists of three main modules: an epidemiological encoder, LLM-based state label generation, and epidemiological representation learning. Experimental validation on three real-world datasets demonstrates that EIN outperforms current state-of-the-art Graph Neural Network methods in terms of both accuracy and robustness. Furthermore, the EIN model effectively manages noise and data sparsity issues across shallow and deep propagation trees, exhibiting notable adaptability. While LLMs help reduce annotation costs, I still question the reliability of using LLMs for accurate label generation. The authors should clarify the advantages of LLM-based labeling in maintaining accuracy while reducing annotation costs.

**Questions:**

However, I have three doubts regarding the approach and its premise:
1.	The authors employ LLMs for state labeling. Given the potential accuracy variance between LLM-generated and human-labeled data, could the authors clarify if LLM-generated labels surpass human labels in accuracy?
2.	Based on Section 4.5.1, the proposed method performs best across datasets when the initial values for infection rates α, β fall within 0 to 0.5. When setting these values randomly, would it be more reasonable to constrain them to a 0 to 0.5 range?
3.	According to the results in Section 4.6, detection accuracy for the "Non-rumor-denial" and "rumor-denial" categories remains relatively stable over time, while "Non-rumor-support" and "rumor-support" categories show considerable fluctuations. Could the authors explain the underlying reasons for these variations?

**Reviewer Confidence:**

4: The reviewer is certain that the evaluation is correct and very familiar with the relevant literature

**Scope:**

3: The work is somewhat relevant to the Web and to the track, and is of narrow interest to a sub-community

---

### Official Review · Reviewer_hRro · 2024-11-21

**Novelty:** 5
**Technical Quality:** 5

**Review:**

Summary：

This paper proposes a new Epidemiology-informed Network (EIN) model to enhance the performance of rumor detection. The model integrates epidemiological knowledge to overcome the sensitivity of existing data-driven methods to data quality. Researchers use large language models to generate user stance labels, avoiding the costly and time-consuming process of manual annotation. Experimental results show that EIN not only outperforms existing methods on real-world datasets but also demonstrates stronger robustness at different tree depths.

Strengths:

1.The logic and structure of the paper are very clear.

2.The idea of integrating epidemiological knowledge into rumor detection is innovative, enhancing the robustness and accuracy of the model.

3.The experiments are very detailed, providing a comprehensive analysis of the model's effectiveness.

Weaknesses:

1.EIN relies on large language models to generate user stance labels. But the labels are not entirely reliable.

2.According to the experimental results in Section 4.4 Ablation Study and Section 4.5.2 Effect of Epidemiology-informed Representation Learning Coefficient, the epidemiology-informed representation learning component have very limited improvement on the Weibo dataset. The paper lacks an analysis of the reasons for this phenomenon.

**Questions:**

This paper mentions that the labels generated by Gemma 2-9B are not entirely accurate. How would switching to a better large language model perform in this case? Currently, API calls for many advanced large language models are quite convenient and affordable.
Additionally, can the application of large language models be further expanded in future research, not just for determining negative or positive, but also for generating more complex labels or further processing text data?

**Reviewer Confidence:**

3: The reviewer is confident but not certain that the evaluation is correct

**Scope:**

3: The work is somewhat relevant to the Web and to the track, and is of narrow interest to a sub-community

---

### Official Review · Reviewer_tWSs · 2024-11-24

**Novelty:** 5
**Technical Quality:** 6

**Review:**

Strengths:

1. The paper provides an in-depth analysis of the research gaps.
2. I appreciate the further discussion in Section 3.4, which helps in understanding the method from a fundamental perspective.
3. The case study is deeply analyzed, allowing for a qualitative understanding of the experimental results.

Weaknesses:

1. The scientific problem is not clearly defined.
2. The methods section is not closely linked to the research gaps.

Comments:

1. The research gap analysis is thorough and supported by experimental evidence. However, I have a question: you mentioned two challenges but did not summarize the fundamental challenge. These two challenges seem to lack a clear connection.
2. The overview diagram is overly complex. I would prefer to see the challenges and innovations of the method highlighted. Detailed implementation can be elaborated upon in the corresponding sections rather than including all details in the overview.
3. Due to the unclear definition of the scientific problem, the methods section lacks a strong connection to the scientific question.

Additionally, I encourage authors to improve these comments as response.

**Questions:**

1. In the introduction, you mention two challenges that lead to noisy data. What type of noise is introduced? Is it related to "more complex propagation trees"? If so, do you have a definition for "complex"? What does "complex" mean in this context?
2. What is the purpose of the chapter "Epidemiological Transmission Model in Rumor Detection"? What key message are you trying to convey to the readers through this analysis?
3. According to the analysis in the third paragraph of the introduction, the scientific problem seems to be related to "complex propagation trees." How is this connected to the methods section?

After reading authors response, most of issues are clear to me now.

**Reviewer Confidence:**

3: The reviewer is confident but not certain that the evaluation is correct

**Scope:**

3: The work is somewhat relevant to the Web and to the track, and is of narrow interest to a sub-community

---

### Official Review · Reviewer_oiNg · 2024-12-02

**Novelty:** 4
**Technical Quality:** 4

**Review:**

This paper proposes an epidemiology-informed network for rumor detection. Particularly, the authors integrate epidemiology theory to improve the robustness of rumor detection caused by data-driven methods. They use large language models to generate stance labels to learn epidemiology-informed representations. The results demonstrate that EIN outperforms the selected methods.

Pros:
1) The authors propose EIN for rumor detection to overcome the data-driven methods' sensitivity to data quality.
2) The experiments show that their framework outperforms existing methods.

Cons:
1）More description of epidemiological states should be given. It is unclear why they integrate epidemiology theory with rumor detection.
2）The improvement of EIN compared with ResGCN and RAGCL is limited.
3) The x-axis of Figure 1 is missing.

**Questions:**

1) Why traditional graph-based methods can not capture capture useful signals from deeper trees?

**Reviewer Confidence:**

3: The reviewer is confident but not certain that the evaluation is correct

**Scope:**

4: The work is relevant to the Web and to the track, and is of broad interest to the community